# FREENETS: LEARNING LAYERFREE NEURAL NETWORK TOPOLOGIES

## ABSTRACT

We propose a novel data driven approach to neural architectures based on information flows in a Neural Connectivity Graph (NCG). This technique gives rise to a category of neural networks that we call "Free Networks", characterized entirely by the edges of an acyclic uni-directional graph. Furthermore, we design a unique, data-informed methodology to systematically prune and augment connections in the proposed architecture during training. We show that any layered feed forward architecture is a subset of the class of Free Networks. Therefore, we propose that our method can produce a class of neural graphs that is a superset of any existing feed-forward networks. Additionally, we demonstrate the existence of certain classes of data, which are expressible through FreeNets, but not through any other feedforward architecture over the same number of neurons. We perform extensive experiments on this new architecture, to visualize the evolution of the neural topology over real world datasets, and showcase its performance alongside comparable baselines.

## 1 INTRODUCTION

Neural architecture search (NAS) has been a widely studied and important problem for many decades now Kitano (1990); Fahlman & Lebiere (1989); Zoph & Le (2016). Various approaches to neural architecture search include reinforcement learning methodologies (Zoph & Le, 2016; Zoph et al., 2018; Pham et al., 2018), techniques based on genetic algorithms (Stanley et al., 2019; Xie & Yuille, 2017; Miikkulainen et al., 2019; Real et al., 2017; 2019), optimization-focused strategies (Luo et al., 2018; Guo et al., 2019b; Xu et al., 2023; Lawton et al., 2023; Shin* et al., 2018) and probabilistic techniques Ru et al. (2020); White et al. (2021). In addition to the above search techniques, heuristic methods (Sapkota & Bhattarai, 2022) and hand-crafted architectures have also been extremely successful (Wang et al., 2023; Cai et al., 2018). Further, such algorithmic approaches for exploring competing neural architectures have emerged as a distinct sub-domain within AutoML, which seeks to automate machine learning pipelines. (He et al., 2021; Jin et al., 2019).

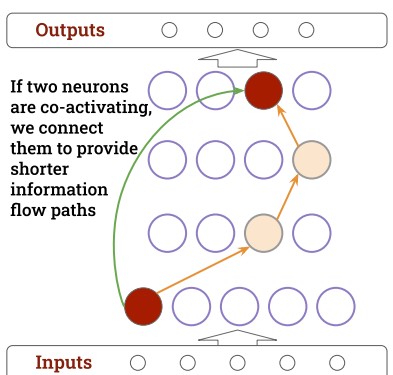

**Augmenting edges using the neural coactivation matrix (NCAM)**

If two neurons are co-activating, we connect them to provide shorter information flow paths

Figure 1: Connecting neurons that capture the same information frees up the path of their co-activation to learn other information.

While varying search strategies have been deployed for NAS, the search space has often been restricted. A frequently employed search strategy involves optimization over a sequence of blocks (Zhang et al., 2022; Chu et al., 2020; Zoph et al., 2018; Liu et al., 2018a;b). Another quintessential example of a restricted search space is chain-structured neural networks (Elsken et al., 2019). The neural blocks – often called as cells, or motifs – are repeated with varying connections between them. The configuration among these cells, albeit optimizable using the above-mentioned techniques, is still not a *free* architecture in terms of allowing arbitrary connections amongst neurons.

We posit that conceptualizing a network purely as a layered sequence of neurons is rather constrictive over the search space of all neural architectures. For example, neurons over a single layer may have interconnections. Such a constraint is not necessitated by the functional structure of feed-forward

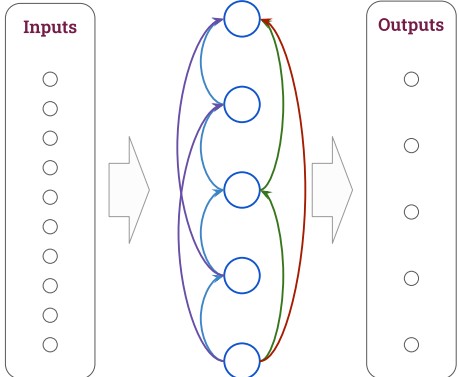

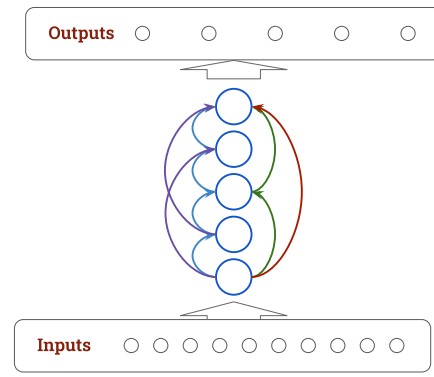

(a) Consider a FREENETS initialization with $k$ neurons. Such an initialization may be viewed as a 2 layer neural network which is fully connected to the inputs and outputs

(b) The FREENETS architecture may also be viewed as consisting of $k$ layers, where there are $k$ neuronal compositions before we get to the outputs. This allows for a much richer function space for the outputs.

Figure 2: Different views of the FREENETS architecture, which is the initialization for our proposed Neural Architecture search method.

and back-propagation. For these, the only requirement is the existence of an acyclic computation graph, which is the same as the existence of a topological order (not necessarily unique) amongst the neurons in the network (Rigo, 2016). In order to use remove such a constraint over neural connections we propose a novel layer-free graph over neurons — an architecture that we call FREENETS.

To initialize the FreeNet architecture, we start with a single parameter: the neuron count within the network. Given such an input, the system yields an initialization characterized by dense interconnections amongst these neurons, adhering to a logical computational sequence. Such a computation sequence is provided by a natural topological ordering amongst the graph nodes. Indeed, the existence of a partial order (a topological order) amongst the nodes in a graph is sufficient to fulfill the acyclicity constraint (Bolte & Pauwels, 2020). This acyclicity, when maintained among the graph nodes, satisfies the essential prerequisites for executing mathematical operations over the network consistently, including back-propagation, which is governed by the chain rule (Dong et al., 2022; Savine, 2019; Schulman et al., 2015). We illustrate such a graph in the Figure 3.

Although this initialization provides a favorable starting point for learning the neural topology, an effective algorithm is requisite to evolve it further. For this we seek an associative learning strategy from neuroscience called Hebbian learning (Wang & Orchard, 2017; Bi & Poo, 2001). Simply put, the Hebb's law states that "Neurons that fire together, wire together". This suggests firstly that neuronal connections in the brain are not static (unlike in traditional neural networks), and further, that augmenting neuronal connections based on data may be effective. While this provides us with a motivation for adding edges within the neural topology from the neuroscience perspective, one may also view it from the lens of information flow. For two neurons from distinct layers exhibiting a strong activation correlation, information is relayed between them through some (potentially long) path within the neural network. Directly connecting these neurons could maintain their activation patterns, thereby liberating preceding neurons in the pathway to assimilate alternative patterns. While such edge additions resemble residual or skip connections, they have not been systematically explored. Analogously, studies on neuroplasticity in children indicate an initial abundance of neural synapses that diminish over time (Mundkur, 2005). This suggests the presence of numerous non-beneficial neuronal connections that may be transmitting spurious or non-helpful information – underscoring the need for a systematic pruning approach.

The neuron activation patterns, as studied by Maini et al. (2023) suggests an interplay between data, weights and neural architecture. Furthermore, several recent works have studied the dynamics of how training impacts activation of neurons in a network (Stephenson et al., 2021; Baldock et al., 2021; Jiang et al., 2020; Feldman & Zhang, 2020). However, these investigations have predomi-

nantly focused on individual neurons rather than information flow. This suggests a potential need to examine the connectivity edges within a neural graph, rather than merely the vertices. To discern the influence of neuronal edges on a network's information flow, we propose an examination of pairwise activation patterns.

For a specified network topology with associated weights, an example might either activate a neuron or leave it dormant. This suggests the representation of an example in the form of a boolean vector within the neural activation domain, denoted as $\boldsymbol{a}_i$ for example $i$. We can study neural co-activations for this example $i$ through the rank-1 matrix $\boldsymbol{a}_i \cdot \boldsymbol{a}_i^\top$. Analogously, we can construct such matrices for the entire dataset and study the co-activation patterns. Such patterns can tell us where to augment edges between neurons in the network topology, in a manner similar to hebbian learning. Further, by studying $\overline{\boldsymbol{a}_i} \cdot \overline{\boldsymbol{a}_i}^\top$, we can understand when to prune neuronal edges in the connectivity graph.

In summary, we suggest an approach to learn data-dependent neural architectures that encompasses existing neural topologies. We start with simple neurons as building blocks, and iteratively learn a computation graph over these neurons to construct a neural network predictor function.

We now lay out the main contributions of this work.

### 1.1 OUR CONTRIBUTIONS

The main contributions in the paper are as follows: 1. We propose a novel, learnable Neural network architecture that we call FREENETS. The architecture learning is based on Hebbian learning principle from neuroscience that says neurons that fire together wire together. 2. The FREENETS framework is capable of learning any feed-forward structure conforming to an acyclic computation graph. Hence, we prove that our approach encompasses the capabilities of conventional feedforward architectures. 3. Our network learns the topology and correspondingly, weights for the topology in a manner similar to alternating minimization. 4. The updates that we perform on the neural topology are not necessarily local. Hence, unlike other neural architecture optimization methods that search over neighbours for a suitable update, it is possible for us to move out of local minima through non-local updates. 5. We prove that there exist functions that can be captured by the FREENETS architecture, and no other feed-forward architecture over $k$ neurons. This shows that the model complexity class of FREENETS is fundamentally different from traditional feed-forward networks.

## 2 RELATED WORK

**NAS Methods:** Neural Architecture Search (NAS) has emerged as a important framework in the realm of neural architecture design. Several approaches to NAS use RL-based methodologies (You et al., 2020; Zoph et al., 2018; Zhong et al., 2018; Zoph & Le, 2016), including Q-learning (Wu & Jain, 2021), Monte Carlo tree search (Wang et al., 2020), and inverse reinforcement learning (Guo et al., 2019a). Liu et al. (2018b) introduced differentiable methods for NAS, which has been successfully employed by various others including Wu et al. (2019); Li et al. (2020); Shu et al. (2022); Wang et al. (2021). Some of these methods also consider pruning (Ding et al., 2022).

Such differential methods entail some computational overhead, and several strategies have been proposed to mitigate these, such as parameter sharing (Pham et al., 2018), predicting accuracy without training (Mellor et al., 2021), by analyzing the spectrum of the neural tangent kernel (Chen et al., 2021), accuracy proxies (Abdelfattah et al., 2021), and neural architecture transfer (Lu et al., 2021).

Though gradient-based techniques are prevalent in architecture search, evolutionary network design presents an alternative and historical paradigm in this domain (Stanley & Miikkulainen, 2002). Several recent works have successfully deployed evolutionary strategies including (Real et al., 2017; 2019; Xie & Yuille, 2017). Some noteworthy evolutionary strategies include predictor assisted strategies Peng et al. (2022) covariance based strategies (Sinha & Chen, 2023) and biologically inspired strategies (Amato et al., 2019).

**NAS Search Space:** A neural architecture can be conceptualized as a macro structure, underpinned by a leitmotif of cells. Many NAS studies, such as Liu et al. (2018b); Zoph et al. (2018); Elsken et al. (2019); Cai et al. (2018), have employed a search space centered around cell design. Liu et al. (2017) consider the search space to be the macro-architecture over these cells. Some studies, like (Gao et al., 2019), focus on task-specific architectures such as graph neural networks. Our work focuses on the macro architecture over neurons, which are the motifs we consider, and is task-agnostic.

**Learning Dynamics in neural networks:** The learning dynamics of neural networks and their interplay with data has been considered in several recent works on model interpretability, including on the long tail of data (Feldman, 2020; Feldman & Zhang, 2020), on the learnability of examples (Frankle et al., 2020) and on the simplicity bias (Shah et al., 2020) of neural networks. Localization of memory in neural architectures has been studied by (Baldock et al., 2021; Stephenson et al., 2021). On the other hand, Maini et al. (2023) propose than memorization of examples by neural networks can not be localized to a few layers, but is often determined by a small set of neurons that may be distributed across layers. Sinitsin et al. (2020) consider modifications to the architecture in order to improve long tail efficiency.

## 3 METHODOLOGY

In this section, we describe the method through which FREENETS is initialized and evolves.

### 3.1 MOTIVATION FOR THE FREENETS ARCHITECTURE

The FREENETS architecture is a dense computation graph over the input. We ensure no cycles by requiring that the initial adjacency matrix for this graph be lower triangular. Note that we do not allow for cycles or self-loops, and hence the diagonal entries of such an adjacency matrix are 0. In fact, we will show that the FREENETS architecture is the most complex function one can learn with $k$ neurons (or $k$ computational nodes). This is a good initialization from which to learn about inter-neuronal interactions as every neuron is free to interact with every other neuron (through the weights) for all points in the dataset.

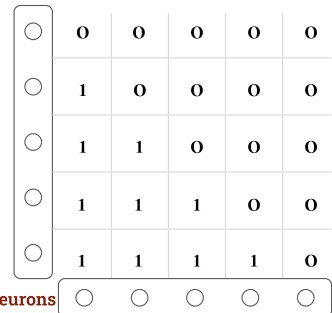

Figure 3: The adjacency matrix for FREENETS initialization is lower triangular, dense and cycle-free.

Traditional architectures have prematurely optimized for the number of weights in the network by handcrafting layer-design. Popular architectures use decreasing layer widths as we move from the inputs to the outputs. Other architectures use an encoder-decoder model which encodes the inputs into a smaller representation space, before decoding these into an output. But we argue that such connections are pre-mature before looking at the data. Indeed, synaptic plasticity is highest at birth and decreases with time Cutler & Mattson (2006). Yet, cognitive function goes up with time, as the same neurons evolve to a sparser connectivity. This neuroscience motivation indicates that beyond requiring a robust initialization, there is also a need for an effective data-dependent learning method to evolve the neuronal topology. We study this question in the next three sub-sections.

### 3.2 "NEURONS THAT ACTIVATE TOGETHER, SHOULD BE WIRED TOGETHER"

While Hebbian learning emphasizes enhancing neuronal connection weights based on synchronous activations, contemporary neural models predominantly rely on backpropagation for weight adjustment. We suggest that both the perspectives are pertinent, but pertain to distinct variables. A neural function encompasses two discrete variable sets: the edges spanning the neurons and the associated weights for the selected edges. Back propagation helps to optimize the set of weights associated with a chosen set of edges in a neural topology. On the other hand, Hebbian learning may be used to adjust this edge set. Indeed, one may seek to optimize over these two distinct objectives, in a manner similar to alternating minimization.

We introduce here the notation of a neural connectivity graph `NCG`, which specifies the edge set between $k$ neurons in the network as a binary adjacency matrix. Note that $NCG \in \{0, 1\}^{k \times k}$.

We can also look at augmenting interneuronal connections from an information flow perspective. Consider two neurons in far away layers that are disconnected, but are firing together for certain examples: see Figure 1. There exists some path where information is flowing to both of these neurons (for the said examples). Such a path may be long, and may employ many other neurons. By directly connecting the neuron that is computed first to the later neuron, we may shorten the

| Neuron 1 Activation | Neuron 2 Activation | $1-2$ Connectivity | Update to Perform | Explanation |
|---|---|---|---|---|
| 0 | 0 | 0 | Do not augment | They are not connected and not firing together, so nothing to do |
| 0 | 0 | 1 | Prune $1-2$ edge | Both are inactive neurons. In the backward pass, the weight cannot be updated. Hence prune. |
| 0 | 1 | 0 | Do not augment | Need not connect. Neuron 2 is being activated by other neurons. |
| 0 | 1 | 1 | Do not prune | Weights can be updated on back-prop. |
| 1 | 0 | 0 | Do not augment | Neuron 1 may be activating other neurons. Further, an added edge-weight may not be updated during back-propagation. |
| 1 | 0 | 1 | Do not prune | They are connected, and not firing together. Yet the first neuron may be inactivating the second neuron. |
| 1 | 1 | 0 | Add $1-2$ edge | We can add an edge to shorten the information flow path. |
| 1 | 1 | 1 | Do not prune | They are connected and firing together, so nothing to do. |

Table 1: Summary of NCG updates based on NCAM and NCIM. Assume that neuron 1 comes before neuron 2 in the forward computation graph.

information flow path. Further, we may use the earlier path to capture other information, thereby expanding the information capacity of the network.

### 3.3 PRUNING NEURONAL CONNECTIONS

While we have looked at interneuronal edge augmentation through hebbian learning, we might ask: "is there any way to prune edges in the NCG"? Consider neurons that are mostly inactive together. Such neurons may co-activate for a small set of examples, and then the information may be passed deeper into the network. But for this small subset of examples, one may argue that since they have already been identified (by the first neuron), this information can be directly passed to the deeper layers, or to the outputs. Such a method suggests that depth in a network is valuable, and hence it should be used for examples occurring more commonly. Sparse examples may simply be memorised by early neurons (and passed to outputs). Hence we suggest that neurons that are jointly inactive but share an edge between them, should have the edge pruned.

Now we look at how to construct statistics regarding such co-activation in the next subsection.

### 3.4 CONSTRUCTION OF THE NCAM AND NCIM

We introduce some notation that will aid in understanding the construction of the neural co-activation matrix NCAM, and neural co-inactivation matrix NCIM. Let the data $\mathcal{D}$ to the neural network come in the form of samples from a space $\mathcal{X} \times \mathcal{Y}$, such that $\mathcal{D} := \{(x_1, y_1), \ldots, (x_n, y_n)\}$ where each $(x_i, y_i) \in \mathcal{X} \times \mathcal{Y}$. Let us consider a neural graph $\mathcal{G}$ consisting of $k$ neurons and $\ell$ edges, with number of weights $\ell$. Let $\text{NN}_\mathcal{G} : \mathcal{X} \to \mathcal{Y}$ be the estimator of the function mapping $\mathcal{X}$ and $\mathcal{Y}$. $\text{NN}_G$ is parameterized with $\ell$ weights $\{w_1, \ldots, w_\ell\}$ where $w_i \in \mathbb{R}$ for all $i \in [\ell]$.

Let $w^* = \arg\min_{w \in \mathbb{R}^\ell} \|y - \text{NN}_\mathcal{G}(x)\|_2^2$ be the optimal set of weights. Now we can define $\text{NN}_\mathcal{G}^*$ as the neural function parameterized by an optimal set of weights. Call $\text{NA}^* : \mathcal{X} \to \{0, 1\}^k$ the neural activation function, where $\text{NA}^*$ is a function of $\text{NN}_\mathcal{G}$ and $w^*$. For ease of notation, we assume we are always working with $w^*$ and write $\text{NA}$ to be the neural activation function.

Now we do a forward pass over the data and record activations of the k neurons. Let $\mathcal{S} \in \{0, 1\}^{k \times n} := \{\text{NA}^*(x_i) \text{ for all } x_i \in \mathcal{D}\}$. Now we are ready to define the NCAM matrix as $\text{NCAM} := 1/n \cdot \mathcal{S} \cdot \mathcal{S}^\top$. Note that the computation of NCAM can be parallelized over the examples as follows. We note: $\text{NCAM} = 1/n \cdot \sum_{i \in [n]} \text{NA}(x_i) \cdot \text{NA}(x_i)^\top$. Analogously, if $\overline{\mathcal{S}} := \neg \mathcal{S}$, then $\text{NCIM} := 1/n \cdot \overline{\mathcal{S}} \cdot \overline{\mathcal{S}}^\top$. Therefore, $\text{NCIM} = 1/n \cdot \sum_{i \in [n]} \overline{\text{NA}(x_i)} \cdot \overline{\text{NA}(x_i)}^\top$ where $\overline{\text{NA}(x_i)} := \neg \text{NA}(x_i)$.

Now that we have recorded the neural co-activations, we are in a position to consider what updates to make to the neural connectivity graph NCG based on these. We can threshold the NCAM and NCIM values at a certain preset values. For each $(i, j)$ pair of neurons we define them to be co-activating if $\text{NCAM}(i, j) > 1 - \epsilon$. Similarly, they are said to be jointly inactive if $\text{NCIM}(i, j) > 1 - \epsilon$. Based on these values, we propose an update table to the neural connectivity graph in Table 1. In a matrix form, these may be written as $\textbf{Aug} := (\text{NCAM} > 1 - \epsilon) \wedge \neg \mathcal{G}$ and $\textbf{Prune} := (\text{NCIM} > 1 - \epsilon) \wedge \mathcal{G}$, where $\textbf{Aug}, \textbf{Prune}$ are the augmenting and pruning indicator matrices respectively. This defines

our update equation to be $\mathcal{G}' = (\mathcal{G} \wedge \neg\mathbf{Prune}) \vee (\mathbf{Aug} \wedge \neg\mathbf{Prune})$. We expand this reasoning into the Algorithms below.

## 4   ALGORITHM AND THEORY

We now propose the algorithms pertaining to FREENETS. Algorithm 1, takes as input a dataset $\mathcal{D}$ and a certain number of neurons $k$. It calls Algorithm 2 for updates to the neural connectivity graph.

---

**Algorithm 1** Find best FREENETS architecture using data

---
1: **Input:**
   (1) Data $\mathcal{D} := \{(x_1, y_1), \ldots, (x_n, y_n)\}$ where each $(x_i, y_i) \in \mathcal{X} \times \mathcal{Y}$.
   (2) Number of neurons: $k$.
2: Initialize $\mathcal{G} := \text{FREENETS}(k)$ for $k$ neurons
3: **while** $\mathcal{G}$ does not converge **do**
4:     $\mathcal{G} := \text{FREEEVOLVE}(\mathcal{D}, \mathcal{G})$
5: **end while**
6: **return** $\mathcal{G}$

---

---

**Algorithm 2** FREEEVOLVE: Update Algorithm for FREENETS using Neural Coactivations

---
1: **Input:**
   (1) Data $\mathcal{D} := \{(x_1, y_1), \ldots, (x_n, y_n)\}$ where each $(x_i, y_i) \in \mathcal{X} \times \mathcal{Y}$.
   (2) A neural topology $\mathcal{G}$ consisting of $k$ neurons and $\ell$ edges.
2: Train the weights of edges in $\mathcal{G}$ until convergence.
   i.e. Let $w^* = \arg\min_{w \in \mathbb{R}^\ell} \|y - \text{NN}_{\mathcal{G}}(x)\|_2^2$ where $\text{NN}_{\mathcal{G}} : \mathcal{X} \to \mathcal{Y}$ is the neural network function over $\mathcal{G}$.
   Call $\text{NA} : \mathcal{X} \to \{0, 1\}^k$ the Neural activation function, where $\text{NA}$ is a function of $\text{NN}_{\mathcal{G}}$ and $w^*$.
3: Do a forward pass over the data and record activations of the k neurons.
   Let $\mathcal{S} \in \{0, 1\}^{k \times n} := \{\text{NA}(x_i) \text{ for all } x_i \in \mathcal{D}\}$
4: Compute $\text{NCAM} := \frac{1}{n}\mathcal{S} \cdot \mathcal{S}^\top$ and $\text{NCIM} := \frac{1}{n}\overline{\mathcal{S}} \cdot \overline{\mathcal{S}}^\top$ where $\overline{\mathcal{S}} := \neg\mathcal{S}$.
   These are easily computed as $\text{NCAM} = \frac{1}{n} \sum_{i \in [n]} \text{NA}(x_i) \cdot \text{NA}(x_i)^\top$ and
   $\text{NCIM} = \frac{1}{n} \sum_{i \in [n]} \overline{\text{NA}(x_i)} \cdot \overline{\text{NA}(x_i)}^\top$ where $\overline{\text{NA}(x_i)} := \neg\text{NA}(x_i)$
5: Compute the augmenting and pruning matrices.
   $\mathbf{Aug} := (\text{NCAM} > 1 - \epsilon) \wedge \neg\mathcal{G}$
   $\mathbf{Prune} := (\text{NCIM} > 1 - \epsilon) \wedge \mathcal{G}$
6: Update the graph $\mathcal{G}$.
   $\mathcal{G}' = (\mathcal{G} \wedge \neg\mathbf{Prune}) \vee (\mathbf{Aug} \wedge \neg\mathbf{Prune})$
7: **return** $\mathcal{G}'$

---

We now state three theorems regarding the Algorithms proposed above.

**Theorem 1.** We make two claim regarding the expressive power of FREENETS — viz. the sufficiency and necessity of FREENETS to express a broad function class of data, when the neural topologies are restricted to only $n$ neurons:

- There exists a general class of functions realizable by FREENETS over $k$ neurons, given compactly supported data.

- Such a class of functions is not realizable by any other neural topology over $k$ neurons.

*Proof Sketch.* Consider the set of all polynomial functions of degree $k$ over the set of positive reals. FREENETS can express this function with no error. Other neural functions over $k$ neurons would not be able to generate all the monomials $\{x^1, \ldots, x^k\}$ in order to generate the polynomial output. We can extend these to a domain which is compact, but appropriately translating the inputs (as bias terms are available during neuronal computation). □

We now make a simple claim regarding pruning and augmentation in the network.

**Claim 4.1.** Consider any consistent[1] neural topology over $n$ neurons. Let $\mathcal{G}$ be the neural graph corresponding to this architecture. Further, let $\mathcal{H}_n$ to be class of neural topologies over $n$ neurons reachable by Algorithm 1 for some dataset. Then $\mathcal{G} \in \mathcal{H}_n$.

---
[1] a consistent neural topology which is causal and allows backpropagation, has an acyclic computation graph

The proof of this claim lies in the fact that every neural architecture is a subset of the edges in FREENETS: See Figure 3. This shows that any layered feed-forward architecture is reachable by Algorithm 1 starting from FREENETS. This leads us to the question of what we can say about the fixed points of the algorithm. We now show some properties of this fixed point.

**Theorem 2.** Let $\mathcal{G}^* = \langle\{1, \ldots, n\}, \mathbf{E}\rangle$ be the output of Algorithm 1 where $\mathbf{E}$ is the set of edges in $\mathcal{G}^*$. Then $\mathbf{Aug} \wedge \neg\mathbf{Prune} \subseteq \mathbf{E}$

*Proof Sketch.* The proof of this claim lies in the characterization of the fixed point. If there are any edges in $\mathbf{Aug} \wedge \neg\mathbf{Prune}$, then such edges are added to $\mathcal{G}^*$, and hence the graph is updated by Algorithm 2. Therefore at convergence, $\mathbf{Aug} \wedge \neg\mathbf{Prune}$ is a sub-graph of $\mathcal{G}^*$. □

## 5 EXPERIMENTS

**Datasets and Models:** We perform experiments on 3 Image Classification datasets, MNIST (Deng, 2012), FashionMNIST (Xiao et al., 2017) and a truncated version of Extended-MNIST (Cohen et al., 2017). We convert the images into pixel vectors to train the proposed architecture as well as the baselines. To evaluate FREENETS, we compare them with fully connected neural networks (FCNNs). For a model with $k$ neurons, we map the data to $k$ dimensions using a fully connected layer. In case of FREENETS, we have interneuronal connections as given by the neural connectivity graph. Finally we convert these $k$ activations, to the class probabilities using a fully connected layer. To ensure fair comparison between FREENETS and FCNN, we add extra neurons to FCNN if the number of parameters defer substantially.

**Model and Training setup:** We build the FREENETS architecture with 3 essential components: an encoder, the FREENET connections, decoder. The encoder weights encode the input vector into a set of neurons, the FREENETS algorithm then, applies the connections between the neurons to produce activations. We use these activation values to calculate $\mathcal{S}$ in the FREEEVOLVE algorithm and modify the architecture. The activations are then decoded using decoder matrices to generate class probabilities. To allow efficient modification of architecture, we store the weights of all connections during training, and mask the weights that are pruned at a particular step. During inference, we propose to store only the weights required for the computation, thereby reducing the FLOPS for computation.

All weight matrices were initialized with a Xavier Initializer (Glorot & Bengio, 2010) while the biases were initialized with a uniform distribution between a fixed bound. We use ReLU activation (Agarap, 1803) for all our experiments to encourage neuron level gating.

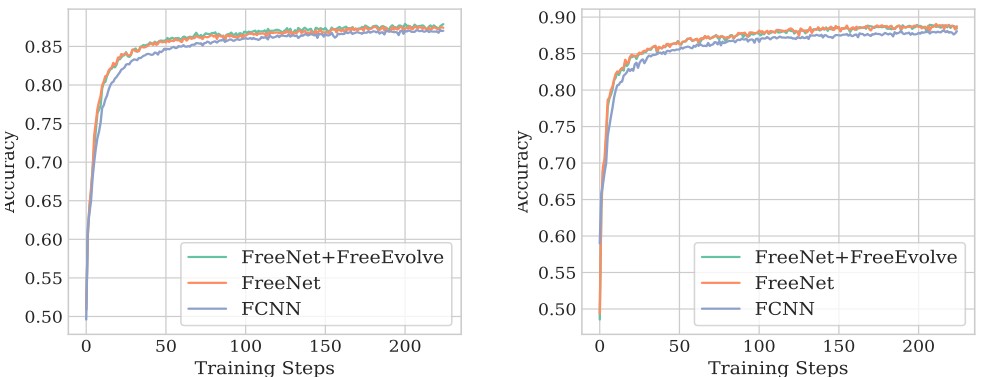

Figure 4: Variation of Validation accuracy on FashionMNIST for architectures with (i) 40 Neurons (ii) 90 Neurons. The Training curve of FREEEVOLVE approximately stays above the FCNN curve.

**Model Training** In the figure 4, we compare the proposed architecture with FCNNs in terms of training convergence. We generally find the best FREENETS outperform the best FCNN with same number of neurons, thereby suggesting the superior connectivity of neurons in the architecture.

**Variation of Accuracy and number of neurons** We compare the best accuracy achieved with the number of neurons present in the architecture, in figure 5. We observe that for FashionMNIST dataset, the performance gap between FREENETS and FCNN increases by adding more neurons to the model. We observe that the MNIST dataset is very easy for neural network with sufficient representative power, and hence all of our approaches converge to a similar value. We can see our improvements more profoundly on the harder datasets - FashionMNIST and EMNIST.

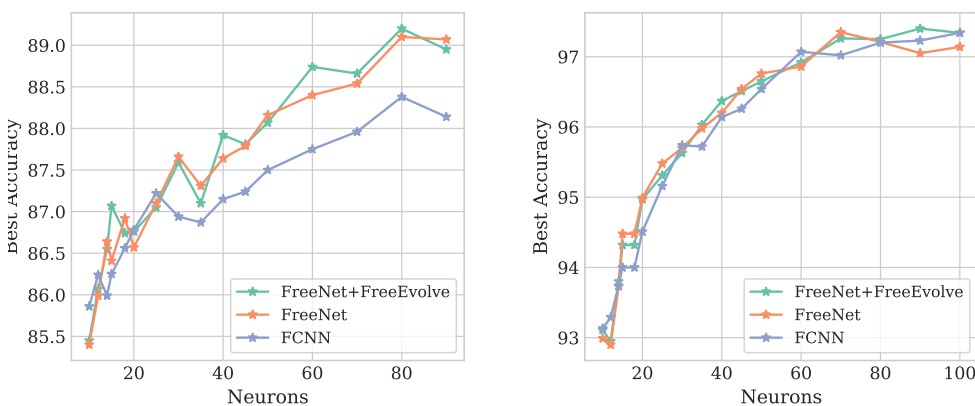

Figure 5: Variation of Accuracy with Number of neurons for (i) FashionMNIST (ii) MNIST

| Dataset | Model | 30 | 35 | 40 | 45 | 50 | 60 | 70 | 80 | 90 | 100 |
|---------|-------|-----|-----|-----|-----|-----|-----|-----|-----|-----|-----|
| MNIST | FREEEVOLVE | 95.63 | 96.03 | 96.37 | 96.51 | 96.65 | 96.92 | 97.26 | 97.25 | 97.4 | 97.34 |
| | FCNN | 95.74 | 95.72 | 96.14 | 96.26 | 96.54 | 97.07 | 97.02 | 97.2 | 97.23 | 97.34 |
| FMNIST | FREEEVOLVE | 87.59 | 87.1 | 87.92 | 87.81 | 88.07 | 88.74 | 88.66 | 89.2 | 88.95 | 88.89 |
| | FCNN | 86.94 | 86.87 | 87.15 | 87.24 | 87.5 | 87.75 | 87.96 | 88.38 | 88.14 | 88.23 |
| EMNIST | FREEEVOLVE | 78.77 | 80.75 | 81.52 | 81.38 | 82.58 | 83.33 | 84.27 | 84.23 | 85.29 | 85.51 |
| | FCNN | 78.55 | 79.97 | 80.63 | 81.92 | 81.7 | 82.63 | 84.07 | 84.12 | 84.6 | 84.98 |

Table 2: Accuracy of each of the Datasets with number of neurons. In each pair, first value represents accuracy using FREEEVOLVE and Second represents standard feedforward with equal parameters.

**Evolution of Neural co-activations during Training** To examine the evolution of different components of the algorithm, we study the variation of number of weights pruned and augmented with Training steps. The plot in figure 6, clearly suggest the convergence of the FREEEVOLVE algorithm. At the start of the training, a significant portion of model weights are pruned. To counter that effect, the augmenting matrix acts after the pruning step to correct the model. Both the Matrices, counter each other's affect and find the model FREENETS architecture. Near the end of training, i.e. When the model has converged the pruning and augmenting matrices stop affecting the model, indicating convergence.

**Prune Matrix and Augment Matrix** We study the distribution of values of these matrices, and how they are affected during training. To generate this plot, we run our training at $\epsilon = 0.25$. The Density values of these matrices are shown in Figure 7.

Through this experiment, we are able to conclude that the model is able to adapt to the pruning and augmenting of connections, thereby producing the activations to satisfy our constraint. The Prune matrix, contains the collection of neurons that are not activating together, and the Augment Matrix contains the collection of neurons that are activating together. By observing that the Prune matrix values are decreasing and the Add matrix values are increasing, we can positively say that we are enhancing the information flow through network.

## 6 CONCLUSION AND FUTURE WORK

In summary, we have proposed a neural architecture search method that we initialize through a graph over individual neurons. The neural architecture evolves through the method of Neural Co-

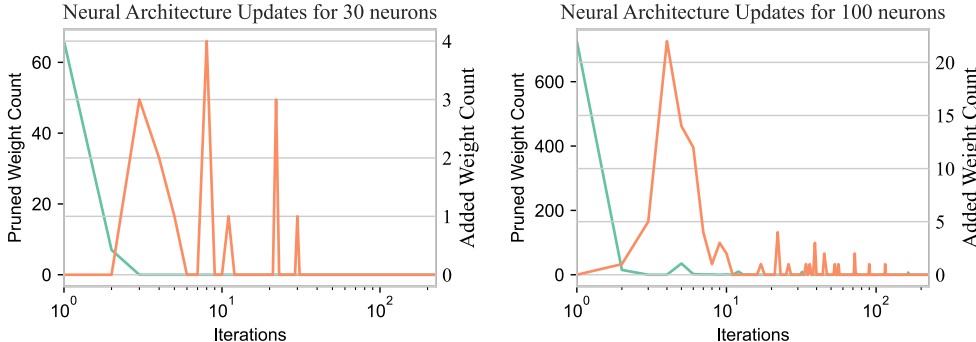

Figure 6: Per-iteration Prune/Add weight count for the model with 30 Neurons (left) and 100 neurons (right).

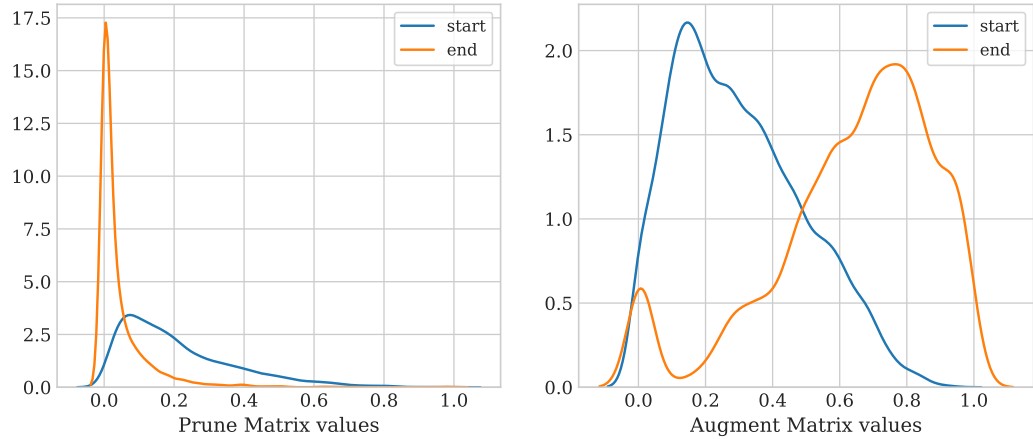

Figure 7: KDE Plots of Prune and Augment matrices at the start and end of the training schedule. The values of the Prune matrix reduce, which implies that the constraints posed by the FREEEVOLVE algorithm are more easily satisfied as the training progresses. The same can be conluded about the Augment matrix

Activation and Neural Co-inactivation (`NCAM` and `NCIM` respectively), which we use to augment and prune neuronal edges. Our method can achieve non-local updates to the neural architecture, which may help it to evade local minima. We contend that our framework embodies a macro neural architecture search approach. It is adaptable to include convolutional cells or RNN blocks based on task requirements. Thus, we assert that the FREENETS framework offers the versatility to accommodate a diverse range of neural functionalities in a neuron efficient manner.

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
