# OpenReview forum: "Freenets: Learning Layerfree Neural Network Topologies"
_ICLR.cc/2024/Conference — ICLR 2024 Conference Withdrawn Submission_

### Official Review · Reviewer_d4y1 · 2023-11-01

**Soundness:** 1 poor
**Presentation:** 3 good
**Contribution:** 2 fair
**Rating:** 3
**Confidence:** 4

**Summary:**

The paper proposes a more flexible architecture than the well known MLP by allowing any two nodes (not just those in adjacent layers) to be connected with a weight. In other words, there is no longer a notion of layers. The authors propose a method to dynamically adjust which nodes are connected based on the alignment of their output. The resulting architecture, called free nets, is evaluated through experiments on MNIST, FashionMNIST and  Extended MNIST.

**Strengths:**

As mentioned in the paper, the new proposed architecture only requires knowing the total number of neurons instead of fixing the architecture. Such architecture can heavily reduce the workload needed to design a good architecture for a new task.

The additional capabilities of the architecture is intuitive but the paper also shows this more rigorously by providing a class of functions that can be represented by freenets and not by fully connected networks.

The paper is relatively easy to follow.

**Weaknesses:**

1. The process of  repeatedly alternating between fine-tuning and pruning phases is widely used for reducing the size of neural networks. There are also several methods (e.g. [1]) that recover pruned weights. As such, this work is basically proposing a new dynamic pruning method based on the firing activity of neurons. A comparison with some of state of the art pruning methods is therefore essential and similarly these methods should be included in the literature review.

2. One of the main obstacles that I can observe is that the paper does not clarify how this model can be adapted to larger scales. In particular there are at least two challenges: a) there is a need for an initial FC layer which maps the input features to freenet neurons. This layer is extremely big for larger number of neurons as the output of this layer has dimension equal to the total number of neurons in the network. b) The freenet starts with all weights connected. Therefore, the total number of weights is much larger than a MLP with multi layers but same number of neurons. This incurs both memory and computation cost especially since this model has to first be trained until convergence (as the first step of free-evolve). Adding neurons increase the cost quadratically instead of linearly in the MLP case.

3. The evaluation is performed on small scale datasets. Such experiments are not at all convincing for a new architecture, especially since as the paper also mentions, it is quite easy to get a high accuracy on such datasets, making it hard to distinguish shortcomings of a method. Indeed the effect of free-evolve is not demonstrated. No confidence interval is also reported.

4. The rules for pruning the weight seem arbitrary. While a rationale is provided for why neurons that do not fire together get disconnected, the same rationale (passing info directly to deeper neurons) could be applied to neurons that fire together to justify disconnecting them. Similarly if neuron a provides a signal to neuron b that prevents it from firing (which could possibly require a non-linear activation to properly work) it is not clear why such connection should be removed.


[1] Lin, Tao, Sebastian U. Stich, Luis Barba, Daniil Dmitriev, and Martin Jaggi. "Dynamic model pruning with feedback." arXiv preprint arXiv:2006.07253 (2020).

**Questions:**

1. You mention that you add more nodes to FCNN if the number of parameters differ "significantly". What does this mean exactly? Can you report what is the exact architecture of the compared FCNN?

2. What is the number of parameters at the end of training (both absolute number and in terms of the percentage of the pruned weight in comparison with the initial complete graph)?

2. How is the threshold $\epsilon$ chosen? Is $\epsilon = 0.25$ used for all experiments?

---

### Official Review · Reviewer_hDot · 2023-11-01

**Soundness:** 1 poor
**Presentation:** 1 poor
**Contribution:** 1 poor
**Rating:** 1
**Confidence:** 2

**Summary:**

The paper presents an approach to neural architecture search by viewing a neural network as a graph of neurons and optimizing their connectivity. The experiments are performed on MNIST, FashionMNIST and EMNIST showing some improvements over fully connected networks.

**Strengths:**

1. The proposed approach (allowing arbitrary connections between neurons) has some potential as indeed typical NAS methods have very restrictive search spaces.
2. The paper includes some formal proofs supporting the claims, although I haven't verified the proofs for correctness.
3. Empirical results show some improvements over fully connected networks and show the ability of the proposed algorithm to prune weights making the model potentially efficient.

**Weaknesses:**

The paper has very limited contributions. Specifically, contribution 1 "The architecture learning is based on Hebbian learning principle from neuroscience that says neurons that fire together wire together." is very questionable. There has been numerous research on the pruning and Hebbian learning topic starting from 1980s and this submission does not properly discuss its relationship to related papers. For example, see the paper "Sparsity in Deep Learning: Pruning and growth for efficient inference and training in neural networks" that has some good references to that. Contributions 2-4 are not convincing, because this submission only compares to a fully connected model and does not compare to other NAS/pruning baselines. There are previous papers like "Graph Structure of Neural Networks" that also considered the connectivity between neurons for NAS/pruning and this submission could discuss it as well.

Experiments are performed on very small scale tasks and the gains over the baseline in the range 0.5-1%. The standard deviation of the results is missing making the comparison harder.

The paper is not clear sometimes and the overall presentation quality requires further polishing (typos, figures partially cut out, etc.).

**Questions:**

none

---

### Official Review · Reviewer_6txU · 2023-11-09

**Soundness:** 3 good
**Presentation:** 4 excellent
**Contribution:** 3 good
**Rating:** 6
**Confidence:** 3

**Summary:**

The paper proposes using a Neural Connectivity Graph (NCG) for neural architecture search, leading to FreeNets. FreeNets are layer-free, making them different from traditional feedforward neural networks. The NCG is initialized as an acyclic, uni-directed graph with dense topological ordering. Then, it alternatively optimizes two architectural designs: the edge sets and their weights. While the weights are updated through backpropagation, the edge sets are pruned and augmented in a data-informed algorithm inspired by learning theories from neuroscience. The authors provide theoretical justification for the expressive power of FreeNets, and evaluate FreeNets empirically on image classification datasets against fully-connected neural networks, showing improvements particularly on more challenging datasets.

**Strengths:**

**[S1]** The paper is very well-written, with clear structure, sound logic, and well-explained motivation. The presentation of the paper is augmented with notations, diagrams, and pseudocode for easy understanding and reproducibility. The related works comprehensively demonstrate how the paper fits the current research context.

**[S2]** The idea of FreeNet is novel in allowing communication amongst all neurons, breaking the layer-wise framework of conventional feedforward neural networks.

**[S3]** The pruning and augmenting strategies using data are well-motivated by Hebbian learning and other theories from neuroscience, which is very interesting, building a connection to information flow among neurons.

**[S4]** The empirical evaluation covers a good range of experiments to study the effects of training steps, number of neurons, counts of evolved weights, and coactivation matrices.

**Weaknesses:**

**[W1]** While the idea of pruning and augmenting pairwise interactions motivated by Hebbian learning is novel and exciting, FreeEvolve does not demonstrate much improvement over vanilla FreeNet from the evaluation (Figure 4 and Figure 5). FreeEvolve is a significant technical contribution of the paper. However, the main improvement seems to come from the initialization from NCG that breaks the layer constraint, but not the evolutionary part. In addition, It may be worthwhile to investigate why a lot more edges are pruned (with a scale of 60 or 600) compared to the number of edges (with a scale of 4 or 20) that are augmented, as shown in Figure 6.

**[W2]** The baselines used for evaluation are only fully-connected neural networks (FCNNs). Although the experiments cover a good range of evaluation, they can significantly enhance the contribution of FreeNets if stronger baselines are also used. For example, neural networks with random residual connection, a randomly initialized NCG with consistent neural topology constraint imposed, or other evolutionary strategies for neural architectural search. Furthermore, while FCNNs match the number of neurons with FreeNets for fair comparison, alternative baselines can also match the number of edges of the produced graph of FreeEvolve.

**[W3]** The computation cost of training FreeNets and FreeEvolve is not adequately discussed in the paper. Each training step of FreeNets+FreeEvolve requires a few iterations of (i) training the weights and (ii) then pruning or augmenting, until convergence. The algorithm can incur very high computational costs compared to training FCNNs, which only requires one iteration of step (i). In the paper, it is unknown how many iterations are required until convergence for Algorithm 1. It would be better to discuss the training cost theoretically and empirically.



**[Minors]**

**[M1]** Some labeling of figures is mismatched. For example, Figure 3 in the second paragraph of Page 2 is not referenced. Also, Figure 6 has a missing legend.

**[M2]** Some typos: Page 2, first paragraph, “In order to use remove”; Page 4, first paragraph, “proposes than”; Page 4, second last paragraph, “NCG” formatting is wrong.

**[M3]** Letter capitalization is not entirely consistent. For example, “Hebbian” and “hebbian” are both used. Additionally, while “Figure *” is used in the first half of the paper, it becomes “figure *” in later sections.

**Questions:**

**[Q1]** Is $\epsilon$ set to 0.25 for all experiments? The value of $\epsilon$ determines which edge to prune and augment, and, therefore can be critical to the model performance. Is there any ablation to study the effect of this hyperparameter?

**[Q2]** For the proof sketch of Theorem 1, is the non-linear activation function taken into account? How will the expressive power of FreeNets compare to fully connected layers with non-linearity?

**[Q3]** What is the exact setup for FCNNs in evaluation? The only information seems to be the number of neurons, but not the number of layers. Furthermore, are the neurons from encoder & decoder for FreeNets also considered when counting the number of neurons?

**[Q4]** The authors claim that FreeEvolve can achieve non-local updates. Is there more justification for this claim?